# Subfractional Spectrum of Serum Lipoproteins and Gut Microbiota Composition in Healthy Individuals

**DOI:** 10.3390/microorganisms9071461

**Published:** 2021-07-08

**Authors:** Daria A. Kashtanova, Natalia S. Klimenko, Olga N. Tkacheva, Irina D. Strazhesko, Victoria A. Metelskaya, Natalia V. Gomyranova, Sergey A. Boytsov

**Affiliations:** 1The “Russian Clinical Research Center for Gerontology” of the Ministry of Healthcare of the Russian Federation, Pirogov Russian National Research Medical University, 16 1st Leonova Str., 129226 Moscow, Russia; tkacheva@rgnkc.ru (O.N.T.); istrazhesko@gmail.com (I.D.S.); 2Center for Precision Genome Editing and Genetic Technologies for Biomedicine, Institute of Gene Biology, Russian Academy of Sciences, Vavilova Str., 34/5, 119334 Moscow, Russia; natasha.klmnk@gmail.com; 3National Research Center for Therapy and Preventive Medicine of the Ministry of Health of the Russian Federation, 10 Petroverigsky Str., 101990 Moscow, Russia; vametelskaya@gmail.com; 4National Medical Research Center of Cardiology of the Ministry of Health of the Russian Federation, 15a Str. 3rd Cherepkovskaya, 121552 Moscow, Russia; NGomyranova@gnicpm.ru (N.V.G.); prof.boytsov@gmail.com (S.A.B.)

**Keywords:** gut microbiota, 16S rRNA sequencing, atherosclerosis, lipoproteins, subfractional analysis of lipoproteins

## Abstract

**Aim:** To reveal the relationship between gut microbiota composition and subfractional spectrum of serum lipoproteins and metabolic markers in healthy individuals from Moscow. **Methods:** The study included 304 participants (104 were men), who underwent thorough preclinical assessment to exclude any chronic disease as well as cardiovascular pathology. Lipoprotein subfractional distribution was analyzed by Lipoprint LDL System (Quantimetrix, Redodno Beach, CA, USA). Gut microbiota composition was assessed by 16S rRNA sequencing of V3-V4 regions. **Results:** High gut microbiota diversity was positively associated with HDL-cholesterol (C) level and negatively associated with abdominal obesity, BMI, and dyslipidemia. According to selbal analysis, excessive representation of *Prevotella* spp. was positively associated with IDL-C and LDL-2-C. VLDL-C correlated with Ruminococcus_u/Faecalibacterium_prausnitzii balance. An unexpected positive relationship between LDL-C level and Bifidobacteriaceae_u/Christensenellaceae_u to Bifidobacterium_u balance was found, which may reflect the importance of the integrative microbiota assessment. Low microbiota diversity was associated with obesity, abdominal obesity and low HDL-C level. **Conclusions:** Gut microbiota imbalance may be one of the components involved in metabolic disorders. The balance of microorganisms and the microbiota diversity may play a more significant role in human health than individual bacterial genera.

## 1. Introduction

Cardiovascular events resulting from the formation of atherosclerotic plaques and atherothrombosis continue to be a heavy burden on the healthcare system and the economy worldwide. Despite all efforts made by scientists and physicians, we still do not have any unambiguous understanding of the mechanisms of the development of cardiovascular diseases. Numerous studies have shown that elevated blood levels of low-density lipoproteins (LDL) cholesterol (C), as well as low levels of high-density lipoproteins (HDL)-C, are associated with an increased risk of developing symptomatic atherosclerosis [1]. Nevertheless, a significant proportion of patients suffering from cardiovascular disease (CVD) have normal or even reduced LDL-C levels and normal HDL-C concentrations. Traditional CVD risk factors, including dyslipidemia, do not always allow to correctly stratify patients according to the level of individual risk of cardiovascular events. Indeed, we can often observe different levels of CVD and different pathways for the development of cardiovascular events in individuals with seemingly similar risk factors according to lipid profile and SCORE scale indicators. Thus, the traditional schemes for determining the risk of CVD developing need further improvement, since the vast majority of coronary events still occur in low and moderate risk groups.

Plasma lipoproteins represent a heterogeneous population of particles varying in density, lipid-protein composition, and size. According to size, LDL are composed of large (LDL1), middle (LDL2), and small dense particles (sdLDL3-7). LDL size seems to be an important predictor of cardiovascular events and progression of CVD. There is increasing evidence that sdLDL are strongly associated with higher CVD risk. A number of studies in recent years have shown that analysis of lipoproteins’ subfractional distribution can provide the additional information concerning their function and role in CVD, both as in the case of HDL [2] and LDL [3]. It is worth noting that the main focus of this article is the LDL fractions.

The gut microbiota could play an active role in lipoprotein metabolism. Epidemiological and preclinical studies showed an association between bacterial communities and cholesterolemia. However, this association remains poorly understood and characterized. Studies investigating the relationship between the composition of the gut microbial community and the concentration of cholesterol transported by lipoproteins have shown that the overrepresentation of certain gut bacteria has a close relationship with blood cholesterol levels, and microbiota transplantation from people with dyslipidemia to mice leads in cholesterol levels increase in these animals [4,5]. Other studies report that use of certain bacterial species, such as probiotics, may decrease host cholesterol levels [6]. The ability of microorganisms to convert cholesterol into coprostanol has long been known. Coprostanol is absorbed much less in the intestine. In recent years, studies have indicated that many previously unknown gut microorganisms are involved in this process [7]. Another interesting aptitude of gut microorganisms is their ability to bind cholesterol directly. For example, some intestinal lactobacilli have this property [8]. The same lactobacilli, enterococci, and some bifidobacteria can deconjugate bile acids—the process, which complicates cholesterol absorption [8,9].

Recently we studied gut microbiota of relatively healthy individuals free of CVD manifestations but who had CVD risk factors. We did not find any relationship between lipoprotein levels and microbiota composition according to standard analyzes of lipoproteins using the homogeneous enzymatic colorimetric method [8,9,10]. For some studied participants, subfractional distribution of apolipoprotein (apo) B-containing lipoproteins was analyzed using by Lipoprint LDL System. The system uses polyacrylamide gel electrophoresis to separate different lipoprotein subfractions by size.

The purpose of the current study was to evaluate the relationship between gut microbiota composition and subfractional profile of serum low densities lipoproteins and their metabolic status in healthy Moscow inhabitants.

## 2. Materials and Methods

### 2.1. Recruitment of Study Participants

The study included Caucasian participants (from Moscow and Moscow Region) aged over 18 years, after the preventive outpatient screening. None of participants had been treated with any drugs (for at least 6 months); nobody had any clinical manifestations of CVD or other chronic diseases. Sex was defined as biological sex. Informed consent was obtained from all subjects involved in the study. Data privacy was ensured by using anonymized identifiers. The study was approved by the local ethics committee, meeting #8, 29 November 2011. Patients’ pre-assessment and exclusion criteria described in detail in the previous paper [10]. Potential participants with any significant deviations in examinations were not included in the study.

### 2.2. Sample Collection and Gut Microbiota Analysis

Stool samples were frozen and stored at −20 °C and thawed before analysis. To assess microbiota composition, sequencing of the 16S rRNA gene variable V3–V4 regions was performed (subsequent to total DNA isolation and library preparation) by using the MiSeq Reagent Kit v2 and MiSeq sequencer (Illumina, San Diego, California, CA, USA) according to Illumina recommendations. Libraries were prepared with ‘16S Metagenomic Sequencing Library Preparation: Preparing 16S Ribosomal RNA Gene Amplicons for the Illumina MiSeq System’ protocol (15044223 Rev. B) using the Nextera XT Index Kit (Illumina) with a dual indexing strategy.

### 2.3. Lipoproteins Subfractional Spectrum Analysis

Lipoprotein subfractional distribution was analyzed by Lipoprint LDL System (Quantimetrix, Manhattan Beach Boulevard Redondo Beach, CA, USA). Separation of lipoprotein particles by size was carried out by 3% polyacrylamide gel electrophoresis (PAGE) in tubes with subsequent scanning and analysis using Lipoware Analysis Program. This method allows to separate lipoproteins subfractions without their preliminary processing with determination the portion of each subfraction (in percent) and the amount of cholesterol (in mg/dl) in the following lipoproteins: very-low-density lipoprotein (VLDL), intermediate-density lipoprotein (IDL) IDL-A, IDL-B, IDL-C, LDL-1, LDL-2, LDL-3, LDL-4, LDL-5, LDL-6, LDL-7, and HDL.

### 2.4. Intima-Media Thickness Assessment

Intima-media thickness (IMT) assessment was carried out using Q-LAB (Philips, The Netherlands) carotid artery duplex scan in B-mode with ECG recording. IMT <0.9 mm was considered to be normal; 0.9–1.3 mm indicated increased thickness. Atherosclerosis was defined as IMT >1.3 mm or a local increase in IMT of 0.5 mm or a 50% increase in nearby IMT. Plaque was considered as a local IMT thickening higher than 1.0 mm, which caused lumen stenosis but did not affect its internal anatomy.

### 2.5. Bioinformatics Analysis

The primary processing of 16S rRNA gene sequencing data was carried out using the Knomics-Biota platform [11] using closed-reference OTU picking with Qiime 1.9 [12] and the GreenGenes [13] reference base with preliminary cropping and filtering of low-quality reads. Alpha diversity was calculated using the chao1 and Shannon metrics. Alpha diversity measures summarize the structure of the ecological community in terms of its number of taxonomic groups as well as its distribution.

### 2.6. Statistical Analysis

The differences of clinical characteristics between groups of patients were assessed using Student’s *t* test. The dependency between intima-media thickness and lipoprotein subfractions was calculated using a linear model. Multiple comparison correction in both analyses was conducted with Benjamini–Hochberg method.

The search for associations between metadata and microbiota composition was carried out on the level of species using the selbal package [13,14]. This package is suitable for working with compositional data, it implements an algorithm that searches for the ratio of bacterial abundances that may predict the factor of interest. The following linear model was analyzed with selbal algorithm: balance ~age + sex + factor (in case of abdominal obesity, which is a binary factor—logistic regression). The selbal algorithm includes cross-validation. We considered reliable the results where the bacteria were selected to the best balance in more than 50% of cross-validation iterations and at the same time the final model R2 > 0.2.

Additionally, the associations between each bacterium and metadata were analyzed using linear model: taxon_abundance ~age + sex + factor. For this analysis taxon abundance was preliminary clr-transformed, zero values were imputed with Bayesian-multiplicative replacement (cmultRepl function from the package zCompositions [15]. The multiple comparison correction was performed on each taxonomic level using Benjamini–Hochberg method.

Associations between alpha-diversity of the microbial community and metadata were assessed using linear model: diversity ~age + sex + factor. The multiple comparison correction was performed using Benjamini–Hochberg method.

## 3. Results

The study included 304 participants aged 52 ± 13 y.o. The clinical characteristics are shown in Table 1. As one can see from this Table, age risk group (men aged ≥ 45 y.o. and women aged ≥ 55) had more risk factors, such as higher body mass index (BMI), systolic blood pressure (SBP), Atherogenic index of plasma (AIP) and glucose level, as well as bigger waist circumference. Men were younger than women. Despite this, with false discovery rate (FDR) adjustment, men were more likely to have some risk factors, which is quite common. In general, we can say that the distribution of risk factors is also quite typical.

We analyzed a total of 12 lipoprotein subfractions, including VLDL, three IDL (IDL-A, IDL-B, and IDL-C) subfractions, seven LDL subfractions and HDL. Levels of lipoproteins (mg/dL) subfractions are presented in Table 2.

Besides HDL-C, which is commonly higher in women, VLDL-C differed significantly between men and women and was much higher in men. It is known that men sex is a risk factor for the onset of cardiovascular events, which is also aggravated by a less favorable profile of cholesterol subfractions. Levels of cholesterol in IDL-B, IDL-A, LDL-2, and total LDL-C were higher in the aged group. All these subfractions can be considered as potentially pro-atherogenic.

As a next step, we analyzed lipoprotein subfractional distribution in subjects who were underwent carotid artery dopplerography by means of linear regression. According to the results, higher VLDL and IDL-B cholesterol levels were associated with intima-media thickening (Table 3). It is noteworthy that none of the other studied subfractions showed their significance not only after the correction for multiple comparisons, but also in the primary analysis.

### 3.1. Gut Microbiota Composition

All samples passed the quality control of classified reads proportion (all samples contained at least 70% of classified reads). Gut microbiota composition was quite similar to that in other studies [16,17,18]. The most presented bacteria were Clostridiales, Ruminococcaceae, Bacteroides and some others (Figure 1). Such composition is normally observed among residents of Western countries, and correlates with the data of literary sources. Detailed interactive visualizations of microbiome characteristics are available here https://biota.knomics.ru/lipids-healthy-individuals (accessed on 19 June 2021).

### 3.2. Association between the Gut Microbiota Composition and Subfractional Spectrum of Apo B-Containing Lipoproteins

First, it should be pointed that there were no associations between gut microbiota composition and lipid spectrum when linear model approach after clr transformation was applied. Another analysis variant was selbal algorithm and it revealed a number of reliable associations (Figure 2): reproducible (>50% of cross-validation iterations included a taxon) and with good prediction quality (R2 of a final model was higher than 0.2) (Table 4). Selbal algorithm operates with microbial balances. Microbial balances are specific microbiome features that allow to perform differential abundance analysis in a compositionality-aware way. Balance is proportional to the log ratio between geometric means of two bacteria groups denoted numerator and denominator [14]. The choice of methods using balances is justified by the fact that the gut microbiota is a complexly organized community. The analysis of individual microbe as a rule is effective only when one microorganism is the causative agent of the disease. On the contrary, balances allow to perform a more comprehensive approach to assessing the relationship between the microbiota and the host health state.

Among clinical parameters, BMI was associated positively with unclassified species from [Prevotella] genus and Enterobacteriaceae family, and negatively—with unclassified genera from Clostridiaceae family. Unclassified species from [Prevotella] genus were also included in reliable balances for IDL-C and LDL-2 as a numerator (positive association). Interestingly, at the same time IDL-C was negatively related to Prevotella copri abundance.

Unclassified species from Christentsenella genus and Bifidobacteriaceae family were positively associated with LDL-C concentrations while other unclassified species from Bifidobacterium genus—negatively. VLDL-C was negatively related to Faecalibacterium prausnitzii abundance and positively—to unclassified species from Ruminococcus genus.

### 3.3. Association between Gut Microbiota Diversity and Metabolic Factors

Remarkable results have been obtained in assessing the diversity of the gut microbiota. Gut microbiota diversity was positively associated with HDL-C level and negatively associated with abdominal obesity, BMI, and AIP (Table 5). Being an extremely important indicator of microbiota well-being, diversity had also been found to be associated with the favorable lipoprotein profile and metabolic health in general.

It should be noted that both low chao and Shannon indexes were positively correlated with obesity as well as abdominal obesity. In contrast, only Shannon index had negative association with AIP and positive—with HDL-C level.

Thus, the bioinformatics analysis revealed the relationship between the general state of the gut microbiota and human health as well as associations of the balance of microorganisms with individual indicators.

## 4. Discussion

Metabolic disorders are the most widespread chronic risk factors in the modern world. To this day, their representation continues to grow, and imbalance in gut microbiota is supposed to be one of the mechanisms for the development of such disorders. In the framework of this study, relationships between gut microbiota with lipid spectrum and some metabolic parameters were found.

Human blood cholesterol level is determined by endogenous cholesterol synthesized mainly in liver and exogenous cholesterol obtained from food components of animal origin. Cholesterol synthesized in hepatocytes is transported to the gallbladder and then secreted into the small intestine along with bile salts. In intestine, biliary cholesterol is mixed with dietary cholesterol, and together they are transported into enterocytes for packaging in chylomicrons and secretion into the blood [15,19]. The role of gut microbiota in these processes is being actively studied, but so far just a few studies have been carried out on the relationship between the results of sequencing of the gut microbiota and lipoprotein subfractions assessment. Serum lipoproteins are represented by a heterogeneous spectrum of particles differing in origin as well as density, size, composition, and functional activity. The main subfractions of lipoproteins are VLDL, IDL (three subfractions C, B, A), large physiologically active LDL-1 and smaller LDL-2 subclasses, small dense LDL-3 to LDL-7, and HDL. Small dense LDL-3-7 subfractions seem to be the most atherogenic, high portion of these particles in the LDL spectrum is associated with the increased risk of atherosclerotic CVD, even if total LDL-C level is normal [20].

In the studied cohort, men had more risk factors than women and age of participants was associated with higher blood pressure, lipids and glucose levels as well as with obesity, including the abdominal obesity. The amount of cholesterol in some lipid fractions such as IDL-B, IDL-A, total LDL and LDL-2 was significantly higher in the age risk group. According to the recent studies, high IDL-C as well as LDL-C concentrations can be associated with coronary atherosclerosis or carotid intima-media thickness. IDL subfractions metabolized to LDL, which are taken up by the LDL receptor in numerous tissues [21]. Cholesterol level in VLDL and IDL-B was also associated with intima-media thickening in our study.

The microbiome composition was associated with metabolic profile. Despite the small group, some bacteria balances were associated with the lipid spectrum. Thus, such unfavorable microbial genera as Enterobacteriaceae_u and poorly characterized [Prevotella]_u were highly represented in obese participants. It is interesting that the same unclassified [Prevotella]_u genus was positively associated with cholesterol content in IDL-C and LDL-2 fractions. Prevotella is quite a common genus in the gut, although it may be harmful due to its potential ability to stimulate inflammation [22]. Role of *Prevotella* spp. within the gut microbiota as well as their effect on the host are not completely understood. Studies show that Prevotella could activate Toll-like receptors, leading to production of inflammatory cytokines [23]. Notably, high representation of Prevotella was found to be associated with insulin-resistance [22], obesity [24], hypertension [25], as well as with non-alcoholic fatty liver disease [26] in case-control studies. In contrast, beneficial bacteria were reduced in those who had high levels of atherogenic lipoproteins. Faecalibacterium_prausnitzii is linked to healthy plant-based foods and produces butyrate—one of the most effective anti-inflammatory agents [26,27].

On the other hand, by means of Selbal analysis we found some unexpected “balances” associated with the lipid spectrum. The balance of Bifidobacteriaceae_u|Christensenellaceae_u and Bifidobacterium_u was associated with total LDL-C level. All these microorganisms are poorly studied, and therefore it is difficult to assess their metabolic profile. Bifidobacteriaceae is a known probiotic taxon. Nevertheless, such association may indicate that the balance of the bifidobacteria community may be no less important than their overall representation. Finally, we have found that the gut microbiota diversity may reflect a normal metabolic status. Low diversity was associated with obesity, abdominal obesity, and dyslipidemia. Our observation that low alpha diversity may correlate with high BMI is consistent with many studies [26,27,28]. Thus, low diversity and dysbalance of the gut microbiota observed in the adult gut may be considered an indicator of metabolic disturbances even in apparently healthy individuals. Low gut microbiota diversity is also considered to be associated with many other conditions, such as inflammatory bowel disease, psoriatic arthritis, diabetes, and arterial stiffness [29].

## 5. Conclusions

The present study has some limitations, the most important of which is the small size of the cohort. It remains imperative to elucidate the routes and mechanisms that may underlie the microbes and host interaction. Nevertheless, we hope that our results could facilitate prospective studies investigating diverse aspects of gut microbiota influence on human health. Although the characterization of “microbial dark matter” still presents serious barriers, the combination of bioinformatics and biochemical approaches may provide access to this largely untapped source of biologically significant metabolic transformations.

## Figures and Tables

**Figure 1 microorganisms-09-01461-f001:**
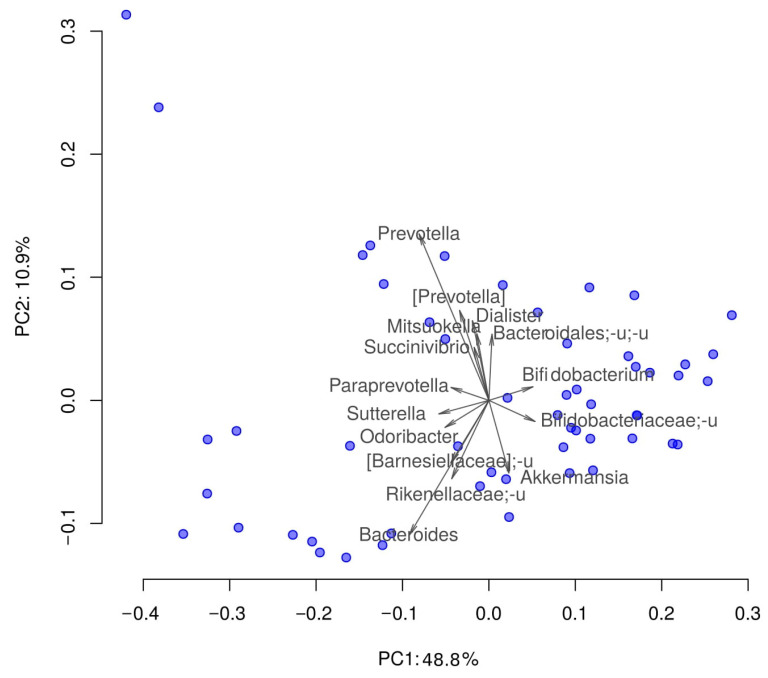
Distribution of the samples by their taxonomic composition in reduced dimensionality. PCoA (Principal Coordinate Analysis); dissimilarity metric: weighted UniFrac. Prefix “_u” denotes all unclassified taxa from the higher taxonomic rank. Arrows show the top 15 taxa in terms of the explained variance in given axes. The arrows’ length is proportional to the percent of variance explained by the taxon. The arrows’ angle reflects the distribution of this variance between the axes.

**Figure 2 microorganisms-09-01461-f002:**
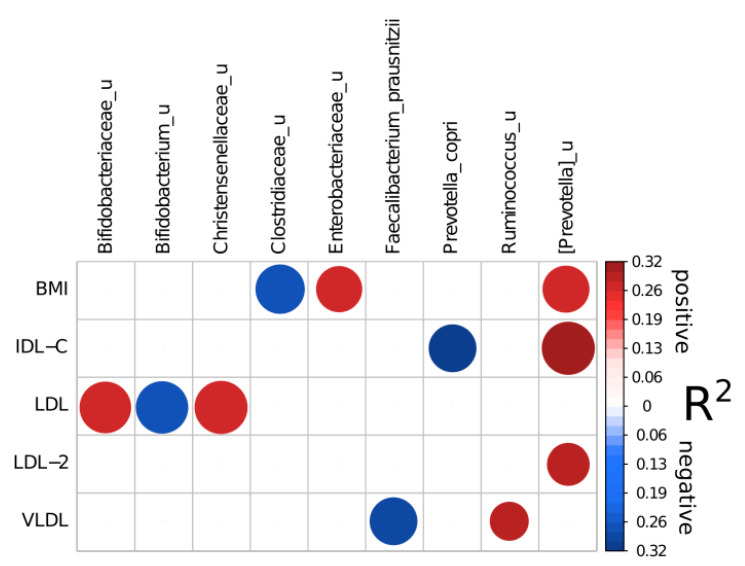
Reliable associations between microbiome composition, subfractional spectrum of apo B-containing lipoproteins and BMI according to Selbal analysis (see Section 2). Circles sizes are related to taxon reproducibility and circles colors to model R2. Numerator parts of balance are colored with red and denominator with blue.

**Table 1 microorganisms-09-01461-t001:** Clinical characteristics of the studied participants.

	Total Cohort (*n* = 304)	Males (*n* = 104)	Females (*n* = 200)	*p*-Value	FDR	Young (*n* = 139)	Age Risk Group (*n* = 165)	*p*-Value	FDR
Age, years	51.5 ± 13.26	48.1 ± 12.03	53.27 ± 13.55	0.0008	0.0014	40.45 ± 8.55	60.82 ± 8.52	-	-
BMI, kg/m^2^	27.36 ± 5.14	28.59 ± 4.22	26.72 ± 5.46	0.0011	0.0017	26.5 ± 5.55	28.08 ± 4.67	0.0086	0.0118
SBP, mmHg	125.33 ± 16.42	129.52 ± 14.61	123.14 ± 16.91	0.0007	0.0014	120.09 ± 15.26	129.74 ± 16.11	<0.00001	<0.00001
DBP, mmHg	78.16 ± 10.26	79.88 ± 10.16	77.26 ± 10.22	0.0342	0.0376	77.24 ± 10.39	78.93 ± 10.12	0.1555	0.1711
Smokers, n	58 (19%)	30 (29%)	28 (14%)	0.0171	0.0210	27 (19%)	31 (19%)	1.0000	1.0000
TC, mmol/L	5.65 ± 1.14	5.61 ± 1.02	5.68 ± 1.21	0.5958	0.5958	5.4 ± 0.97	5.86 ± 1.24	0.0003	0.0005
TG, mmol/L	1.28 ± 0.85	1.6 ± 1.07	1.12 ± 0.65	<0.00001	0.0001	1.09 ± 0.68	1.45 ± 0.94	0.0001	0.0002
HDL-C, mmol/L	1.22 ± 0.31	1.03 ± 0.27	1.32 ± 0.29	<0.00001	<0.00001	1.25 ± 0.31	1.19 ± 0.32	0.1078	0.1318
AIP	−0.07 ± 0.72	0.31 ± 0.72	−0.27 ± 0.64	<0.00001	<0.00001	−0.27 ± 0.71	0.09 ± 0.7	<0.00001	<0.00001
Fasting plasma glucose, mmol/L	5.75 ± 1.4	6.1 ± 1.61	5.56 ± 1.24	0.0036	0.0049	5.3 ± 0.94	6.12 ± 1.6	<0.00001	<0.00001
WC, cm	89.54 ± 15.3	98.67 ± 12.46	84.84 ± 14.5	<0.00001	<0.00001	85.17 ± 15.01	93.21 ± 14.59	<0.00001	<0.00001

Note: Data are mean ± standard deviation. BMI—body mass index, SBP—systolic blood pressure, DBP—diastolic blood pressure, TC—total cholesterol, TG—triglycerides, WC—waist circumference. AIP—Atherogenic index of plasma, logarithmically transformed ratio of molar concentrations of TG to HDL-C. Young—men aged < 45 y.o. and women aged < 55, age risk group—men aged ≥ 45 y.o. and women aged ≥ 55.

**Table 2 microorganisms-09-01461-t002:** Lipoproteins subfractions (mg/dl), sex and age.

Cholesterol, mg/dL	Total Cohort (*n* = 304)	Males (*n* = 104)	Females (*n* = 200)	*p*-Value	FDR	Young (*n* = 139)	Age Risk Group (*n* = 165)	*p*-Value	FDR
VLDL	15.49 ± 4.44	17.14 ± 4.10	14.68 ± 4.39	0.0002	0.0012	14.87 ± 4.29	16.36 ± 4.53	0.0223	0.0580
IDL-C	23.08 ± 8.32	23.49 ± 7.66	22.88 ± 8.65	0.6225	0.6744	22.38 ± 8.17	24.08 ± 8.48	0.1662	0.2700
IDL-B	18.92 ± 5.88	18.89 ± 4.34	18.93 ± 6.53	0.9644	0.9644	17.39 ± 4.86	21.08 ± 6.52	<0.00001	0.0002
IDL-A	21.34 ± 8.09	20.38 ± 8.47	21.82 ± 7.89	0.2569	0.4564	20.04 ± 7.77	23.18 ± 8.22	0.0084	0.0273
LDL-1	39.77 ± 13.7	38.08 ± 12.68	40.61 ± 14.15	0.2109	0.4564	38 ± 11.84	42.27 ± 15.71	0.0417	0.0902
LDL-2	18.8 ± 12.27	21.17 ± 11.99	17.63 ± 12.28	0.0573	0.2484	16.2 ± 11.23	22.48 ± 12.79	0.0006	0.0024
LDL-3	3.65 ± 5.00	4.61 ± 5.82	3.17 ± 4.49	0.0853	0.2773	3.19 ± 4.88	4.3 ± 5.12	0.1310	0.2434
LDL-4	0.61 ± 2.15	0.86 ± 3.29	0.49 ± 1.24	0.3862	0.4564	0.58 ± 2.58	0.65 ± 1.33	0.8172	0.8172
LDL-5	0.08 ± 0.88	0.20 ± 1.50	0.02 ± 0.18	0.3234	0.4564	0.12 ± 1.14	0.01 ± 0.11	0.3055	0.3776
LDL-6	0.01 ± 0.14	0.03 ± 0.25	0.00 ± 0.00	0.3211	0.4564	0.02 ± 0.19	0 ± 0	0.3195	0.3776
LDL-7	0.21 ± 2.40	0.03 ± 0.25	0.30 ± 2.93	0.2984	0.4564	0.02 ± 0.19	0.49 ± 3.71	0.2618	0.3776
Total LDL	127.23 ± 34.74	130.20 ± 27.56	125.75 ± 37.84	0.3551	0.4564	118.26 ± 29.49	140.08 ± 37.72	<0.00001	0.0002
HDL	56.16 ± 15.3	47.83 ± 13.21	60.29 ± 14.61	<0.00001	<0.00001	56.94 ± 14.19	55.05 ± 16.78	0.4135	0.4480

Note: Young—men aged < 45 y.o. and women aged < 55, Age risk group—men aged ≥ 45 y.o. and women aged ≥ 55.

**Table 3 microorganisms-09-01461-t003:** Cholesterol content in lipoprotein subfractions and intima-media thickness (N = 287).

Cholesterol, mg/dL	Linear Regression Coefficient	*p*-Value	FDR Adjusted *p*-Value
VLDL	0.0136	<0.00001	0.0001
IDL-C	0.0035	0.0324	0.0789
IDL-B	0.0080	0.0004	0.0025
IDL-A	0.0016	0.3564	0.4633
LDL-1	−0.0011	0.2864	0.4137
LDL-2	0.0025	0.0237	0.0769
LDL-3	0.0056	0.0364	0.0789
LDL-4	0.0089	0.1543	0.2508
LDL-5	−0.0022	0.8844	0.8844
LDL-6	−0.0282	0.7638	0.8275
LDL-7	0.0098	0.0801	0.1487
Total LDL	0.0009	0.0173	0.0751
HDL	−0.0004	0.6894	0.8147

**Table 4 microorganisms-09-01461-t004:** Relationship between the gut microbiota composition, subfractional spectrum of serum apo-B containing lipoproteins and BMI, selbal test results.

Factor	Full Balance	Adjusted R^2^	Reproducible Taxon	Direction of Association	Percent of Times Included in a Balance
BMI	[Prevotella]_u | Enterobacteriaceae_u)/Clostridiaceae_u	0.26515	[Prevotella]_u	+	72.5
			Enterobacteriaceae_u	+	70
			Clostridiaceae_u	-	80
LDL	Bifidobacteriaceae_u|Christensenellaceae_u/Bifidobacterium_u	0.26381	Bifidobacteriaceae_u	+	87.5
			Christensenellaceae_u	+	92.5
			Bifidobacterium_u	-	90
VLDL	Ruminococcus_u/Faecalibacterium_prausnitzii	0.29299	Ruminococcus_u	+	50
			Faecalibacterium_prausnitzii	-	75
IDL-C	[Prevotella]_u/Prevotella_copri	0.32018	[Prevotella]_u	+	92.5
			Prevotella_copri	-	75
LDL-2	[Prevotella]_u/[Ruminococcus]_gnavus	0.28927	[Prevotella]_u	+	60

Note: Postfix “_u” denotes all unclassified taxa from the higher taxonomic rank. Note: We considered significant only those taxa from the full balances for which a high reproducibility was observed (>50%).

**Table 5 microorganisms-09-01461-t005:** Gut microbiota diversity association with clinical parameters.

	*p*-Value	Linear Model Coefficient	FDR	Alpha Diversity Metric
BMI	0.0017	−17.5031	0.0250	Chao1
Abdominal obesity	0.0083	−194.5565	0.0622	Chao1
BMI	0.0029	−0.0344	0.0147	Shannon
Abdominal obesity	0.0018	−0.3968	0.0138	Shannon
AIP	0.0015	−0.3065	0.0138	Shannon
HDL-C	0.0050	0.6334	0.0187	Shannon

## Data Availability

The data presented in this study are openly available in https://biota.knomics.ru/lipids-healthy-individuals (accessed on 19 June 2021).

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
