# Peer review of "Subfractional Spectrum of Serum Lipoproteins and Gut Microbiota Composition in Healthy Individuals"

_microorganisms, 2021, doi:10.3390/microorganisms9071461_

Round 1
Reviewer 1 Report
In this article, Kashtanova et al. analyzed the relationship of the gut microbiota composition with metabolic markers especially serum lipoproteins. They observed the specific compositional balance which had a relationship with metabolic markers. This study field has a great importance, and it should be regarded as a significance of the study that they tried to link the compositional balance of the gut microbiota to serum lipoproteins in subfractional level. In general, the manuscript is well written and covers attractive topics for readers. However, the manuscript has several points to be addressed before being accepted.
- The overall writing of the manuscript, including grammar, syntax, and word choice, needs considerable improvement. It is critical that the manuscript will be reviewed by a native English speaker.
- Were the subjects screened based on biological sex (sex: female/male)? The words meaning social sex (gender; woman/man) were mixedly used.
- Table 1: 95% CI is not displayed although the footnote says it is described.
- Line 5 in Results section: “older” shall be read as “younger”.
- Line 8 in Results section: What is “an adult population”?
- Line 2 in subsection 3.1.: The author mentioned that gut microbiota composition was quite regular. I wonder how they defined “regular”. It’s not scientific.
- Line 5 in subsection 3.1.: Authors should cite appropriate references to the data of literary sources.
- Figure 1: The plots were distributed horizontally along x-axis. I wonder which microbiota components had effects on PC1, since all the components displayed were directing along PC2.
- Table 4: I cannot find the data of R. gnavus which was related with LDL-2.
Author Response
Dear Reviewer,
Thank you for taking the time to review our work. We greatly appreciate your thoughtful comments that helped improve the manuscript.
- Thank you for the commenе. We’ve corrected errors in the text.
- Yes, the subjects were screened based on biological sex, we’ve srated it in the text.
3, 4, 5, 6, 7. Thank you very much for your suggestions. We’ve amended the text according to your comments.
- Thank you for the suggestion. The figure showed 5 arrows with the highest projection length on the PC1-PC2 plane. According to your suggestion we’ve added 10 more arrows and now some of them show taxa gradient along PC1. We’ve used 15 arrows because adding more arrows made the figure too noisy. We also extended Figure legend to describe the meanings of arrows.
- We considered the association significant if two conditions were met in selbal results:
- good prediction quality of the most frequently chosen balance (R2 > 0.2)
- >50% of cross-validation iterations chose a taxon to be a part of a balance
Second column of Table 4 contains information about all balances for which we observed good prediction quality. But only reproducible taxa from the balances were considered to be significantly associated with metadata. [Ruminococcus] gnavus was included in balance [Prevotella]_u / [Ruminococcus]_gnavus with R2=0.28, but only [Prevotella]_u has reproducibility > 50% (as can be seen in column 4 of the Table 4), so only this taxon was mentioned in the Results and Discussion.
Thank you for highlighting that it is not clearly explained in our manuscript. We’ve added a Note to Table 4 in order to clarify the difference between the members of full balances and significant results.
Reviewer 2 Report
In the present study, the authors aimed to reveal the relationship between gut microbiota composition and subfractional spectrum of serum lipoproteins and metabolic markers in a healthy individuals' group.
Overall, the study is well designed and the results seem promising. Indeed, as the authors mention, the study has enrolled a limited number of patients and the conclusions have to be treated with caution.
To overcome this issue, I would suggest a greater comparison of the results with other studies (more references should be included), from other healthy and patients groups
Finally, All vertical and horizontal lines presented in all tables should be removed, and only the first row should be separated with a horizontal line
Author Response
Dear Reviewer,
Thank you very much for reading our manuscript and providing your valuable comments.
- Thank you for highlighting this. We’ve added more information to compare results with other studies.
- Thank you for your suggestion. We’ve changed the table format.
Reviewer 3 Report
I have read the manuscript entitled "Gut microbiota composition and subfractional spectrum of serum lipoproteins in healthy individuals" proposed by Kashtanova et al.
There are some major issues that should be addressed before further processing of the manuscript:
Introduction: the authors did not provide any general information on gut microbiota, therefore I suggest citing two recent articles that cover most of the basic knowledge on gut microbiota: DOI-1: https://doi.org/10.1152/ajpgi.00161.2019 / DOI-2: https://doi.org/10.3390/jcm9113705.
Materials and Methods: the authors should be more specific in terms of inclusion and exclusion criteria. Also, were data studied for normal distribution? Why employing linear models to study correlations between continuous and discrete variables.
Results/Discussion: in my opinion much information is provided in terms of patients lipid profile with less focus in terms of gut microbiota. Should the authors think of reducing this part in the results or changing the title/aim?
Author Response
Dear Reviewer,
Thank you for your valuable suggestions. They really have helped to improve our manuscript.
- Thank you for advising these articles, we’ve listed one of them in the text.
- We’ve added more specific information and also we’ve cited a work with the previously described criteria.
Thank you for the question. According to assumptions of a linear regression response variable (in our case - microbial characteristics) must be normally distributed for each fixed value of the predictors. In our study, among the tested microbial characteristics there were bacterial abundances and alpha diversity.
- Alpha diversity usually follows a normal distribution. Following your comment we additionally proved it on our data with the Shapiro-Wilks test (shapiro.test R function, p value > 0.1).
- Microbial abundances are a more complex case. This data is initially zero-inflated, discrete, and also compositional. When we calculate microbial balances the aim is to address these issues and make the data more normal. However, we can’t check the normality of all microbial balances that exist, because there are plenty of them. Selbal algorithm provides a procedure to choose one best microbial balance without testing all variants of balances. According to current best practices linear models are allowed for microbiome data after ILR transformation (which is implemented in selbal):
“After the ilr* is applied, standard statistical tools, such as multivariate response, linear regression and classification, can effectively test for differences in the balances or log ratios between microorganisms rather than the raw microbial abundances, controlling for compositionally.” (Knight et al. 2018)
*Ilr - procedure of obtaining balances, selbal implements ilr transformation
Although linear model are not ideal solution for differential abundance analysis, it allow to adjust for covariates (in our case age and sex). In the studies searching for the associations between microbiome composition and metadata linear regression are often used both in case of continuous and categorical predictors as they may be simultaneously included in one mode as covariates. Here are some examples: (Zhernakova et al. 2016) https://science.sciencemag.org/content/sci/suppl/2016/04/27/352.6285.565.DC1/Zhernakova.SM.pdf (Bolte et al. 2021) https://gut.bmj.com/content/gutjnl/70/7/1287.full.pdf(Byrd et al. 2021) https://rupress.org/jem/article-pdf/218/1/e20200606/1405371/jem_20200606.pdf
So, linear regression models are very often used for differential abundance analysis. However, in the case of selbal algorithm, logistic regression is implemented for the binary factors rather than linear regression. Therefore, for the ‘abdominal obesity’ factor (the only categorical factor in our dataset) in case of selbal algorithm logistic regression was used. We are grateful that you brought this to our attention, and corrected the Methods section to make this clear.
- Thank you for this point. We’ve swapped these terms in the title.
Round 2
Reviewer 1 Report
I confirmed that the authors properly addressed my comments raised in the review cycle. Now I can recommend to accept this manuscript for publication.